# Janus Kinase Inhibitors: A New Tool for the Treatment of Axial Spondyloarthritis

**DOI:** 10.3390/ijms24021027

**Published:** 2023-01-05

**Authors:** Marino Paroli, Rosalba Caccavale, Maria Pia Paroli, Luca Spadea, Daniele Accapezzato

**Affiliations:** 1Division of Clinical Immunology, Department of Clinical, Anesthesiologic and Cardiovascular Sciences, Sapienza University of Rome, 00185 Rome, Italy; 2Eye Clinic, Department of Sense Organs, Sapienza University of Rome, 00185 Rome, Italy; 3Post Graduate School of Public Health, University of Siena, 53100 Siena, Italy

**Keywords:** Janus kinases, JAK inhibitors, axial spondyloarthritis

## Abstract

Axial spondyloarthritis (axSpA) is a chronic inflammatory disease involving the spine, peripheral joints, and entheses. This condition causes stiffness, pain, and significant limitation of movement. In recent years, several effective therapies have become available based on the use of biologics that selectively block cytokines involved in the pathogenesis of the disease, such as tumor necrosis factor-α (TNFα), interleukin (IL)-17, and IL-23. However, a significant number of patients show an inadequate response to treatment. Over 10 years ago, small synthetic molecules capable of blocking the activity of Janus kinases (JAK) were introduced in the therapy of rheumatoid arthritis. Subsequently, their indication extended to the treatment of other inflammatory rheumatic diseases. The purpose of this review is to discuss the efficacy and safety of these molecules in axSpA therapy.

## 1. Introduction

Spondyloarthritis (SpA) is a family of chronic rheumatic diseases that share many clinical and pathogenic features. A main characteristic of SpA is the involvement of the spine and sacroiliac joints in the inflammatory process [1,2]. SpA includes ankylosing spondylitis, psoriatic arthritis, reactive arthritis, enteropathic arthritis, and undifferentiated spondyloarthritis [3]. More recently, the term axial spondyloarthritis (axSpA) has been coined. AxSpA comprises a radiographic form (r-axSpA), previously referred to as ankylosing spondylitis, and a non-radiographic form (nr-axSpA). The latter is characterized by sacroiliitis, whose signs are not detectable by conventional radiography but only by MRI of the sacroiliac joints [4]. The Assessment of SpondyloArthritis International Society (ASAS) has identified a set of criteria for the diagnosis of both radiographic and non-radiographic axSpa [4]. These include the presence of a major radiologic criterion, such as radiographic changes in the sacroiliac joints or the presence of bone edema on MRI, plus a minor criterion or a major criterion, such as the presence of the HLA-B27 allele [5] plus two minor criteria. AxSpA is believed to originate from an inflammatory process at the level of the enthesis that can subsequently extend to the joint structures. Joint manifestations may be associated with extra-articular involvement, including the presence of psoriasis, uveitis, or chronic inflammatory bowel disease, particularly Crohn’s disease. Another feature of spondyloarthritis is the presence of dactylitis, characterized by severe inflammation of the finger or toe tendons [6]. All these non-joint conditions are, in many cases, the most important pathological components of axSpA. Traditional therapy of axSpA has been based on the use of nonsteroidal anti-inflammatory drugs (NSAIDs) [7]; however, new innovative drugs have become available in the past 20 years. These drugs, known as biologics, are monoclonal antibodies or soluble receptors capable of blocking with high specificity several cytokines involved in disease pathogenesis. The main targets of these drugs are tumor necrosis factor-α (TNFα), interleukin (IL)-17, and IL-23 [8]. Although the use of these new drugs has greatly improved the quality of life of patients and can slow the radiological progression of the disease, a significant number of subjects do not respond or partially respond to therapy. Therefore, the availability of new drugs for these difficult-to-treat patients is urgently needed [9,10]. In recent years, thanks to intensive research in rheumatic diseases, a new class of drugs called Janus kinase inhibitors (JAKinibs) have become available. These drugs were initially used successfully in rheumatoid arthritis [11]. Later their indications were extended to the treatment of other inflammatory conditions [12,13,14]. The purpose of this review is to report the main data available in the literature on the efficacy and safety of JAKinibs in the treatment of axSpa, which led to the recent approval of some members of this class of drugs for the treatment of the radiographic and non-radiographic form of the disease.

## 2. The Pathogenesis of axSpA

The most likely hypothesis to explain the occurrence of axSpA is that the effect of environmental factors together with a genetic predisposition may induce a chronic inflammatory response involving both innate and adaptive immunity [15,16,17]. AxSpA is strongly associated with the presence of the HLA-B27 allele, which is very rare in the general population [18,19,20,21]. However, the mechanism by which this allele may contribute to the pathogenesis of axSpA is still the subject of intensive studies. Other risk factors include polymorphisms in *RUNX3*, *TBX21*, and *ERAP1* genes [22,23,24]. Mechanical stress at the level of the entheses seems to play an important role as a trigger for the onset of axSpA. In this regard, there is multiple evidence that entheses experimentally subjected to mechanical stress can induce the efflux of innate immune cells from the peri-entheseal tissue directly into the enthesis through transcortical blood vessels [25]. This eventually leads to enthesophyte formation due to the activation of osteoblasts [26]. It is noteworthy that a characteristic target of inflammation in axSpA is the sacroiliac joint, which shares with the entheses the presence of abundant fibrocartilaginous tissue [27]. Alterations in the gastrointestinal microbiota have long been considered a key element in the pathogenesis of the disease, probably through a mechanism of molecular mimicry [28,29,30,31,32,33]. In addition, intestinal dysbiosis, combined with impaired gut barrier function, may allow pathogenic bacteria to invade the intestinal lumen, inducing IL-17 production by cells of both the adaptive and innate immune systems [34]. Recent studies have been devoted to the role of cytokines involved in the pathogenesis of axSpA. Clinical experiences on the efficacy of blocking TNFα, IL-23, and IL-17 indirectly demonstrated the key role of these molecules in the pathogenesis of the disease [35,36,37,38,39]. It should be noted that the function of these cytokines depends on the activation of JAK molecules that mediate their intracellular signaling after the recognition of their respective receptors. It is likely that a therapeutic agent that can inhibit multiple cytokines involved in the pathogenesis of axSpA may be more effective than drugs such as biologics that block a single agent. There is abundant evidence that JAK plays a key role in the genesis of axSpA. Gene mutation studies have shown that polymorphisms in JAK2, TYK2, and STAT3 are associated with the genesis of ankylosing spondylitis [40,41]. The role of JAK in the genesis of axSpA is further supported by several animal models. While biologic drugs are effective in treating axSpA through specific blockade of individual cytokines, particularly TNFα, IL-23, and IL-17A [36,37,42,43], JAKinibs appear to be an ideal therapy because of their ability to block multiple cytokines simultaneously [44]. For all these reasons, JAKinibs have attracted the attention of researchers for their possible use in the therapy of this disease.

## 3. The Function of JAK Molecules

After recognizing their receptor on the target cell, cytokines send their signal to the nucleus through biochemical interactions between different molecules in the cytoplasm. JAK combination with signal transducer and activator of transcription (STAT) is one of the most important cytokine signal transduction pathways [45,46,47]. The JAK–STAT pathway involves those cytokines that bind to type I/II cytokine receptors [48]. There are four members of the JAK family, namely JAK1, JAK2, JAK3, and tyrosine kinase 2 (TYK2). Each cytokine receptor is associated with a homodimeric or heterodimeric pair of these JAK molecules [49]. After the cytokines bind to their receptors, JAK molecules associated with the intracellular portion of the receptor undergo autophosphorylation and then phosphorylate, in turn, the receptor tail on tyrosine [50]. Seven members of the STAT family have been identified, namely STAT1, STAT2, STAT3, STAT4, STAT5A, STAT5B, and STAT6. Dimers of STAT bind to phosphorylated docking sites on receptors, where they are phosphorylated by JAK. Once phosphorylated, STAT molecules dissociate from receptors, form homo- or heterodimers, and migrate into the nucleus regulating the expression of target genes [48,49,50,51,52]. Regulation of gene expression requires the recruitment of coactivators by STAT dimers. These coactivators interact with histone proteins with which nuclear DNA is associated, making specific regions of DNA more accessible to STAT and the nuclear transcriptional machinery [53,54]. STAT molecules are then de-phosphorylated and once dissociated from DNA leave the nucleus. As pointed out earlier, JAK molecules are responsible for the signaling of several cytokines. JAK1, in combination with JAK3, is involved in the signaling of cytokine that recognizes receptors formed by common γ chain (γc), such as IL-2, IL-4, IL-7, IL-9, IL-7, and IL-21 [48,49]. These cytokines are involved in the growth/maturation of lymphoid cells and the differentiation/homeostasis of T cells and natural killer cells [13,44,55,56,57]. IL-7, in particular, modulates innate lymphoid cells (ILC), which are strongly implicated in the pathogenesis of axSpA through production of Il-17 [58]. JAK2 in homodimeric form is associated with receptors recognized by growth factors, including erythropoietin and granulocyte colony-stimulating factor (G-CSF), and with β-chain (βc) receptors recognized by IL-3 and IL-5. Homodimers of JAK2 regulating signaling downstream of erythropoietin and G-CSF play a key role in erythropoiesis and myelopoiesis [59,60]. Importantly, GM-CSF has recently been linked to the pathogenesis of axSpA [61]. TYK2, in combination with JAK2 and JAK1, may be associated with receptors that share gp130 molecules recognized by different cytokines, including IL-6 and IL-11. IL-6 is also involved in the activation of ILC [13,46,59,60,62,63]. TYK2, in combination with JAK1/JAK2, is associated with type I and type II interferon receptors. It is noteworthy that JAK2 and TYK2 regulate IL-12 and IL-23 signaling. The latter cytokine is required to maintain the differentiation state of T-helper 17 cells, which are mainly involved in axSpA immunopathogenesis [64]. Figure 1 shows the JAK molecules associated with cytokine receptors and the subsequent activation of the different STAT molecules. Therefore, it can be assumed, as outlined above, that JAK inhibition affecting the signaling of multiple cytokines involved in the pathogenesis of axSpA can be particularly effective in the therapy of the disease [56,65]. It should be noted that TNFα, which has an established role in the pathogenesis of axSpA, does not recognize receptors associated with the JAK–STAT pathway. However, IL-12 signals through the JAK2-TYK2 pathway together with IFN-γ via JAK1-JAK2, are essential for TNFα production by macrophages [13]. Therefore, blocking JAK2/TYK2 or JAK1/JAK2 indirectly modulates TNFα production through inhibition of IL-12 and IFN-γ production [13].

## 4. Rationale for JAKinib Treatment of axSpA

Biologic drugs are highly effective in the treatment of axSpA. However, a significant number of patients do not respond to the inhibition of anti-TNFα therapy or have secondary failure to such therapy [66]. Anti-IL-17 monoclonal antibodies, while effective in treating patients with axSpa, are significantly less effective in those patients who have been previously treated with anti-TNFα [67]. Therefore, there is an urgent need for new treatments for these difficult-to-treat patients. It has been shown that a large number of cytokines recognize multiple receptors that can induce intracytoplasmic signaling through the activation of different JAK molecules. The cytokines most implicated in the genesis of spondyloarthritis are IL-23 and IL-17. These two cytokines are so functionally connected that the term “IL-23/IL-17 axis” has been coined. IL-23 can maintain the differentiative state of T helper-17 (Th17) cells, impeding these cells’ trans-differentiation into T-regulatory cells (Treg) and inducing Th17 cells to produce IL-17 family members [68]. Specifically, IL-17A and IL-17F mediate tissue damage by stimulating target cells that express receptors for IL-17. These cells, in turn, produce potent soluble proinflammatory factors. This leads to joint erosion, enthesitis, and disorders of bone proliferation [69,70]. In addition to Th17 cells, other cell types belonging to both adaptive and innate immunity can produce IL-17, and thus, participate in the pathogenesis of spondyloarthritis. Among them are IL17^+^CD8^+^ T cells [61,71,72], γδT cells [73,74], mucosal-associated invariant T (MAIT) cells [75,76], invariant natural killer T (iNKT) cells [77], and group 3 innate lymphocyte cells (ILC3) [78]. Four JAK inhibitors are currently available for rheumatic diseases, namely tofacitinib, baricitinib, upadacitinib, and filgotinib. Table 1 shows the indications approved to date for these JAKinib. Baricitinib, a JAK2 inhibitor, is not currently approved or studied for axSpA therapy and has rheumatoid arthritis as its only rheumatologic indication. JAK1/JAK3 selectivity appears to be more effective for the therapy of axSpA. The selectivity of different JAK has been evaluated in vitro through several laboratory tests, including biochemical assays using recombinant JAK molecules and cellular assays in which cell lines are treated with JAK inhibitors and then stimulated with cytokines to assess their ability to prevent phosphorylation of STAT molecules. In these assays, tofacitinib demonstrated preferential inhibition of JAK1 and JAK3, with 5- to 100-fold selectivity over JAK2 [79]. Filgotinib demonstrated 30-fold selectivity for JAK1 vs. JAK2-dependent signaling. Upadacitinib showed higher selectivity for JAK1 than for JAK2, JAK3, and TYK2, demonstrating 60-fold selectivity for JAK1 vs. JAK2 and >100- vs. JAK3 in cellular assays [80]. Table 2 summarizes the JAK selectivity of the approved JAKinibs. Additional JAK/TYK inhibitors are currently under development, but no clinical data have been published to date. It is important to note that the relative selectivity of different JAK inhibitors may vary depending on the assay used [63,81,82,83,84]. Moreover, it should be emphasized that the in vitro tests used may not reflect in vivo concentrations and effects in humans [81,85]. There are numerous differences among JAK inhibitors, in addition to their selectivity, such as chemical structure, inhibition potency, metabolism, and excretion profiles. These variables indicate that the clinical effect of JAKinibs may show significant clinical differences. Consistent with the mode of action of JAKinibs, biomarker analyses have shown that tofacitinib inhibits preferentially JAK1 and JAK3 but also, in part, JAK2 inhibiting the broadest array of cytokines as compared with other JAKinibs. Upadacitinib exerts direct inhibitory activity on several JAK1-dependent factors (IFN-α/β, IFN-γ, IL-6, IL-2, IL-5, and IL-7) and indirectly on several JAK1-independent pathways (IL-1, IL-23, IL-17, IL-18, and TNFα [63] resulting in inhibition of key cytokine-induced events, such as leukocyte activation and mobility, inflammatory response, and connective tissue damage. Filgotinib has also been shown to reduce circulating proinflammatory cytokines and chemokines, adhesion molecules, and markers of matrix remodeling associated with axSpA [86]. In addition, preclinical models have demonstrated the impact of the JAK–STAT blockade on PsA manifestations [87,88] also through a TNF-dependent mechanism and TNF-independent mechanism [87]. As discussed earlier, a single JAK inhibitor allows simultaneous inhibition of multiple cytokines, with possibly greater efficacy. While baricitinib, as mentioned above, has been approved exclusively for rheumatoid arthritis, tofacitinib, upadacitinib, and filgotinib have also been approved for the treatment of psoriatic arthritis and ulcerative colitis. Tofacitinib and upadacitinib have recently received approval for the treatment of rx-SpA, and the indication of upadacitinib has also been extended to nr-ax-SpA based on the results of the SELECT-AXIS 2 study, as explained in more detail below [89]. Therefore, the following section will discuss the different clinical trials aimed at establishing the efficacy and safety of different JAK inhibitors in the treatment of axSpA. Figure 2 shows the immunopathology of axSpA and the point of action of JAK inhibitors.

## 5. JAKinib Clinical Efficacy in axSpA

As discussed above, tofacitinib is a relatively nonselective JAKinib that can mainly inhibit JAK3, JAK2, and JAK1 [52,90]. In a phase II study, tofacitinib proved effective in a small sample of patients with axSpA [91]. This 12-week study looked at 207 r-axSpA patients who received different doses of tofacitinib, 2 to 10 mg twice daily or placebo. The primary endpoint was ASAS20 (Assessment in Ankylosing Spondylitis 20% improvement) response rate at week 12. Patients treated with 5 mg twice daily achieved an ASAS20 response rate of 80.5%, significantly higher than that of the control group (41.2%). Secondary endpoints such as ASAS40 and Bath ankylosing spondylitis disease activity index 50 (BASDAI50), meaning 50% improvement in BASDAI compared with baseline, as well as a change in the ankylosing spondylitis disease activity score (ASDAS), showed a significant improvement with tofacitinib 5 and 10 mg twice daily compared with placebo. Patients with objective signs of inflammation (elevated CRP or spine edema on MRI) of the sacroiliac joint presented greater treatment efficacy compared with placebo. Changes in MRI scores were analyzed at week 12, with a significantly greater reduction from baseline with tofacitinib 5 and 10 mg compared with placebo. Adverse events were similar between treatment groups [91]. This study was later expanded by a phase three study, which enrolled patients with r-axSpA and an inadequate response to at least two NSAIDs [92]. A total of 269 patients were randomized to receive tofacitinib 5 mg twice daily or placebo for 16 weeks, followed by an open-label period with tofacitinib until week 48. The primary endpoint was the percentage of ASAS20 at week 16. Overall, at week 16, there was a greater percentage of ASAS20 response in the tofacitinib group than in the placebo group (56.4% vs. 29.4%). ASAS40 response, universally considered a highly significant secondary endpoint, was greater with tofacitinib than with placebo (40.6% vs. 12.5%). Treatment efficacy was maintained until week 48 [93]. Regarding upadacitinib, it was evaluated in r-axSpA in the SELECT-Axis 1 study [94]. This study was a phase III, randomized, double-blind, placebo-controlled trial. The first part was a 14-week placebo-controlled study, followed by an open-label period of another 14 weeks. This period was further extended with an open-label period. The 187 patients with r-axSpA who were naive to treatment with biologic drugs and had not responded satisfactorily to at least two NSAIDs were included. The patients were then randomized to receive upadacitinib 15 mg daily or placebo for 14 weeks. Patients who completed the first period were admitted to the second phase of the study, where they received upadacitinib open-label until week 104. The primary endpoint for the first part of the study was the response to ASAS40 at week 14. The response was significantly greater in the upadacitinib group than in the placebo group (52% vs. 26%). Several secondary endpoints were also achieved in the upadacitinib group, but not in patients who received placebo. Secondary endpoints included improvement in ASDAS scores, spine MRI radiology score, and percentage of patients with BASDAI50 and ASAS partial remission. The interim analysis of the SELECT-Axis 1 extension study reported efficacy and safety data at 1 year [95]. Results showed sustained treatment efficacy for 1 year and increased ASAS40 response throughout the study. Study. The percentage of patients who responded to ASAS40 was higher at week 64 than at week 14. Patients who switched from placebo to upadacitinib showed a similar level of response to those initially randomized to upadacitinib. No significant side effects occurred in one year. The SELECT-AXIS 2 study examined the efficacy of upadacitinib in nr-axSpA [89]. In this multicenter, randomized, double-blind, placebo-controlled, phase 3 study, adult subjects with objective MRI-based signs of inflammation or elevated C-reactive proteins and an inadequate response to NSAIDs were enrolled. Patients were randomly assigned to receive upadacitinib 15 mg orally once daily or placebo, based on MRI of sacroiliac joint inflammation and high-sensitivity C-reactive protein screening status and previous exposure to disease-modifying biologics. The primary endpoint was the proportion of patients with an ASAS40 response at week 14. Out of a total of 313 patients, 156 received upadacitinib and 157 received placebo. A significantly higher ASAS40 response rate was observed with upadacitinib compared with placebo at week (45% vs. 23%). The rate of adverse events up to week 14 was similar in the upadacitinib group and the placebo group. The conclusion of this study, which led to the approval of upadacitinib for the treatment of nr-axSpA, is that the drug significantly improved the signs and symptoms of active disease compared with placebo [89].

Finally, filgotinib, a selective JAK1 inhibitor [96], was evaluated in patients with axSpA in a double-blind, placebo-controlled phase-two study (TORTUGA) [97]. In this study, patients had active r-axSpA with inadequate response to at least two NSAIDs. Patients included in the study could also have been treated with unsatisfactory results with anti-TNFα biologics. The primary endpoint of the study was a significant change in ASDAS score from baseline. A total of 161 patients were then randomized to be treated with filgotinib 200 mg daily or placebo for 12 weeks. The primary endpoint was met, with greater improvement in ASDAS score at week 12 in the filgotinib group than in the control group. Secondary endpoints, which included improvements in ASAS20, ASAS40, at least 20 percent improvement in function, pain, inflammation, global patient, CRP, and spinal mobility, ASAS partial remission, and functional ankylosing spondylitis index (BASFI) at week 12, were achieved in the filgotinib arm compared with placebo. There was also a significant reduction in inflammation scores on MRI of the spine and sacroiliac joint. Safety was considered satisfactory [97]. Although this analysis showed a greater reduction in inflammatory segments of the spine evaluated, no improvement in bone erosion or new bone formation was observed [98]. On the other hand, the sacroiliac joint study showed a significant reduction in erosion score in the filgotinib group compared with placebo, demonstrating significant drug activity at this level observable as early as 12 weeks [99]. However, despite these promising results, filgotinib has not yet received approval for the treatment of axSpa. Table 3 summarizes the main features and results obtained from the major studies of JAKinib in axSpA therapy. In Figure 3, a flow chart for selecting biologics and JAKinibs for patients with axSpa is provided, in accordance with ASAS-EULAR recommendations [100].

## 6. Safety Issue

The long-term safety of drugs used in chronic disease is a very important issue. Although in clinical trials the safety profile of JAKinib has been satisfactory, the presence of signs of adverse events has emerged in phase IIIb and IV studies in rheumatoid arthritis, where the number of patients treated are much larger than in other rheumatic diseases. To clarify the safety of JAKinib, the FDA requested the ORAL study, a prospective phase 3B/4 safety study in patients with RA to assess the risk of cardiac events, cancer, and infections. The primary objective of this study was to evaluate the safety of tofacitinib at two dosages (5 mg twice daily and 10 mg twice daily) compared with biologics anti-TNFα (etanercept or adalimumab) in subjects with rheumatoid arthritis (RA) with inadequate response to methotrexate, aged 50 years or older, and with at least one additional cardiovascular risk factor. The primary endpoints of this study were determining the risk of venous thromboembolic events, major acute cardiovascular events (MACE), and established malignancies. The study was supposed to include at least 1500 subjects to be followed for 3 years. In total, 4362 subjects received the study treatments. The results showed that the criteria of non-inferiority to anti-TNFα were not met. Moreover, these risks were associated with the dose of 5 mg twice daily. The most frequently reported MACE was myocardial infarction, while the most frequently reported neoplasm was lung cancer. In addition, there was an increased incidence of herpes zoster virus (HZV) infection in patients treated with JAKinib as compared with the control group [101]. However, it should be emphasized that the study was not powered to compare individual dose groups of tofacitinib and anti-TNFα about safety outcomes. Other criticisms of the study were the open-label design, geographic differences in the anti-TNFα used, and the fact that the 10 mg twice-daily dose was withdrawn during the study period [101]. Despite the limitations of the ORAL study, FDA has added a black box warning on the use of JAKinib, including tofacitinib, baricitinib and upadacitinib, for the treatment of rheumatoid arthritis and other inflammatory diseases, emphasizing the increased risk of serious cardiovascular events, neoplasia, and thrombosis. Manufacturers of these drugs have been requested to provide additional safety data through post-marketing surveillance programs. The current lack of extensive post-marketing safety data on JAKinib presents a challenge for physicians in the use of these drugs in assessing the risk-benefit ratio [46]. The still-uncertain safety data have led to the drafting of several guidelines. For example, tofacitinib has been indicated for the treatment of r-axSpA only as a last line of therapy after failure of anti-TNFα and anti-IL-17 and if NSAIDs are not effective in controlling pain symptoms. The only exception to these limitations is the presence of ulcerative colitis, this comorbidity being an additional indication for treatment with tofacitinib. The European League Against Rheumatisms (EULAR) issued additional recommendations, underlying that a thorough clinical history should be taken before starting treatment with JAKinib to ascertain the possible presence of neoplasms, intestinal diverticula, or previous thromboembolic events [11,102,103,104,105]. Particular emphasis was placed on the need to consider the patient’s age and the possible presence of various comorbidities such as diabetes, chronic respiratory disease, and corticosteroid use as potential factors for serious adverse events. EULAR also stressed the need for a complete blood count, assessment of liver and kidney function, serologic testing for HBV and HCV infection, and a search for possible latent tuberculosis. It has also been recommended an annual skin examination for the early detection of skin cancer. A limit of neutrophil and lymphocyte counts below which there is a contraindication to initiating therapy has also been established. The European Medicine Agency (EMA) emphasized the recommendation not to use tofacitinib in patients older than 65 years. Finally, tofacitinib and upadacitinib were not recommended for use in patients with cirrhosis with Child-Pug C score, and the use tofacitinib but not upadacitinib was contraindicated in the case of moderate to severe renal dysfunction. However, because clinical trials on the use of JAKinib in axSpA have shown favorable results in major disease domains, these drugs still deserve important consideration in the therapeutic management of axSpA. However, the prescription of these drugs cannot go without proper communication and agreement with the patient according to the principles of good clinical practice [106], tailoring the treatment to the specific characteristics of the individual patient.

## 7. Conclusions

JAKinibs represent a very attractive therapeutic strategy for the treatment of axSpA. Studies completed to date have included patients with both r-axSpA and nr-AxSpA. Tofacitinib, upadacitinib, and filgotinib have shown improvement in major disease domains and other important domains such as quality of life and fatigue. Less convincing was the response on other disease domains, such as incidence of uveitis or changes in enthesitis score compared with placebo. However, tofacitinib and upadacitinib have gained approval from international regulatory agencies (tofacitinib for rx-axSpA and upadacitinib for both rx-axSpA and nr-axSpA). Further studies are needed in patients who have not responded adequately to anti-TNFα and/or anti-IL-17 biologics to better understand in which line of treatment these drugs should be optimally used. Although safety issues have not yet been fully clarified, it is certainly interesting and reassuring to note that during the clinical development of JAKinibs there was no evidence of increased incidence of extra-articular manifestations such as uveitis, psoriasis, and inflammatory bowel disease compared with placebo. Another important point that needs to be stressed is to ascertain if use of JAKinib in axSpA can block or at least slow the radiographic progression of the disease. Despite all these important considerations and the need for future large randomized controlled trials, JAK inhibitors represent, to date, an innovative therapy for the treatment of patients with axSpA who have failed all other available treatments.

## Figures and Tables

**Figure 1 ijms-24-01027-f001:**
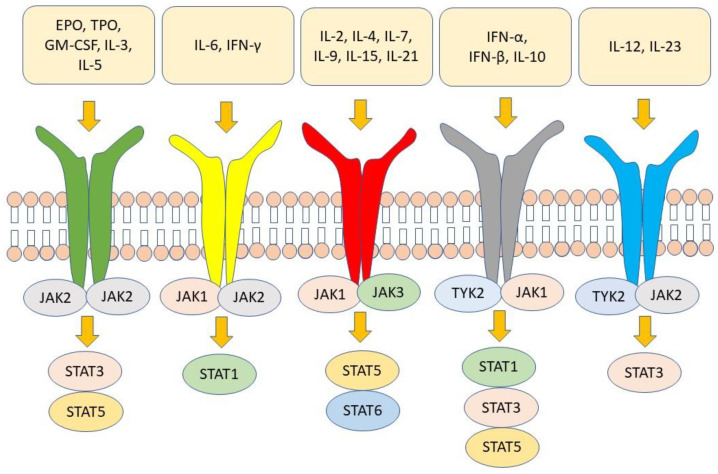
Several cytokines and growth factors recognize receptors that send their signals through the JAK–STAT pathway. EPO = erythropoietin; TPO = thrombopoietin; GM-CSF = granulocyte–macrophage colony-stimulating factor.

**Figure 2 ijms-24-01027-f002:**
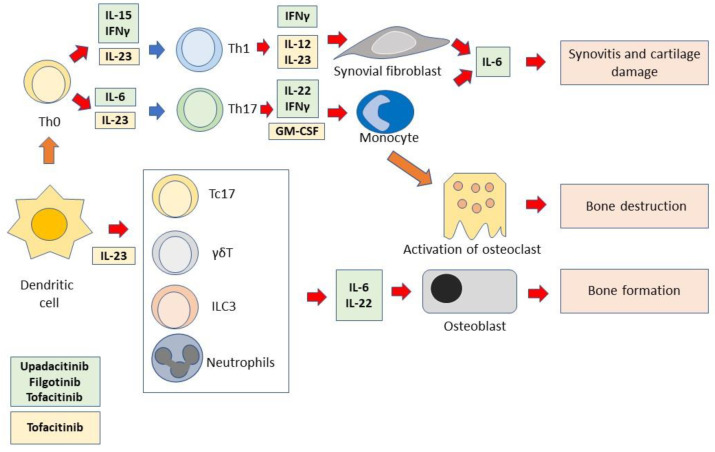
Immunopathology of axSpA and point of action of JAK inhibitors. Dendritic cells induce Th0 cells to differentiate into Th1 or Th17 cells. Dendritic cells also activate different cells of the innate immune system. These functions require the presence of several cytokines. In turn, differentiated/activated immune cells produce cytokines that activate monocytes, synoviocytes, osteoclasts, and osteoblasts. All these cells are responsible for the immunopathological events that cause axSpA. The cytokines depicted in the figure are those that can be inhibited by JAKinibs. The light green box includes the JAK1/JAK3-dependent cytokines that are inhibited by both the selective JAK1 inhibitors, upadacitinib and filgotinib, and the pan-JAK inhibitor, tofacitinib. The light yellow box includes the JAK2/TYK2-dependent cytokines that are inhibited by tofacitinib.

**Figure 3 ijms-24-01027-f003:**
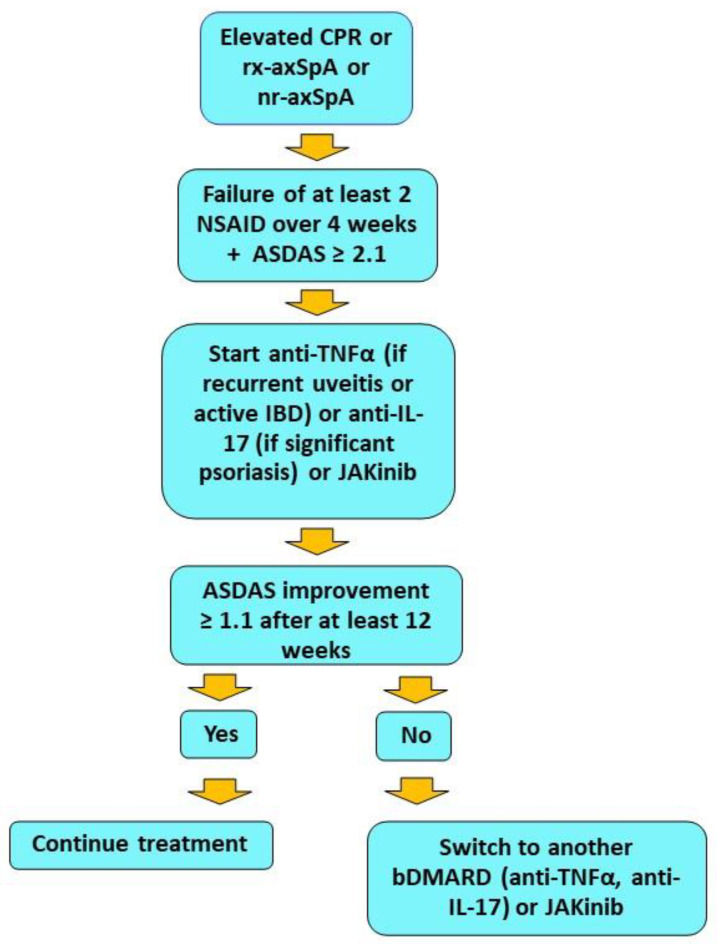
Flow chart showing therapeutic management of axSpA with predominant axial manifestations. The data are in accordance with ASAS-EULAR recommendations. JAK inhibitors, although included as phase II treatment along with biologics, are currently used as the second choice in clinical practice. A positive rheumatologist’s opinion is also required.

**Table 1 ijms-24-01027-t001:** Approved indication of JAKinib in inflammatory diseases.

JAKinib	RA	PsA	UC	r-axSpA	nr-axSpa
Baricinib	Yes	No	No	No	No
Filogotinib	Yes	Yes	Yes	No	No
Tofacitinib	Yes	Yes	Yes	Yes	No
Upadacitinib	Yes	Yes	Yes	Yes	Yes

RA = rheumatoid arthritis; PsA = psoriatic arthritis; UC = ulcerative colitis; r-axSpA = radiographic-axial spondyloarthritis; nr-axSpA = nonradiographic-axial spondyloarthritis.

**Table 2 ijms-24-01027-t002:** Selectivity of JAKinibs.

JAKinib	JAK1	JAK2	JAK3
Baricinib	++++	++++	+
Filogotinib	++++	+	+
Tofacitinib	++	+++	++++
Upadacitinib	++++	++	+

Data are presented in arbitrary semi-quantitative form to summarize the main findings extracted from the literature. ++++ = high; +++ = medium; ++ = low; + = very low.

**Table 3 ijms-24-01027-t003:** Results from clinical studies on JAKinibs in axSpA.

Disease	Study	JAKinib	Primary Endpoint	ASAS40 Response Difference Versus Placebo	Reference
r-axSpA		Tofacitinib	ASAS20 at week 16	27%	[92]
r-axSpA	SELECT-AXIS 1	Upadacitinib	ASAS 40 at week 14	26%	[94]
r-axSpA	TORTUGA	Filgotinib	Change of ASDAS from baseline at week 12	19%	[97]
nr-axSpA	SELECT-AXIS 2	Upadacitinib	ASAS40 at week 14	22%	[89]

## Data Availability

Not applicable.

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
