# Peer review of "Janus Kinase Inhibitors: A New Tool for the Treatment of Axial Spondyloarthritis"

_ijms, 2023, doi:10.3390/ijms24021027_

Round 1

Reviewer 1 Report

Review report form

Article title: Janus kinase inhibitors: a new tool for the treatment of axial spondyloarthritis

The authors here discuss the significance and efficacy of the Janus Kinase (JAK) Inhibitors in the treatment of Axial spondyloarthritis (axSpA). The article is well written and clearly summarizes the efficacy of JAK inhibitors as an efficient treatment for patients with axSpA. This article will definitely help the readers to gain more insights and understanding in targeting JAK as an therapeutic approach. However, the following points need to be considered for improvement of the article.

1.      The authors should site the following articles: (Toussirot E. The Use of Janus Kinase Inhibitors in Axial Spondyloarthritis: Current Insights. Pharmaceuticals (Basel). 2022 Feb 22;15(3):270. doi: 10.3390/ph15030270.) and (Akkoc N, Khan MA. JAK Inhibitors for Axial Spondyloarthritis: What does the Future Hold? Curr Rheumatol Rep. 2021 Apr 28;23(6):34. doi: 10.1007/s11926-021-01001-1.)

2.      The table 1 and table 2 provides least information. The authors should consider adding more data on drug efficacy values, side effects and references.

3.         The clinical trials so far conducted on using JAK Inhibitors can also be depicted in a graphical or tabular format.

Author Response

Reviewer #1

  1. Query: The authors should site the following articles: (Toussirot E. The Use of Janus Kinase Inhibitors in Axial Spondyloarthritis: Current Insights. Pharmaceuticals (Basel). 2022 Feb 22;15(3):270. doi: 10.3390/ph15030270.) and (Akkoc N, Khan MA. JAK Inhibitors for Axial Spondyloarthritis: What does the Future Hold? Curr Rheumatol Rep. 2021 Apr 28;23(6):34. doi: 10.1007/s11926-021-01001-1.). Answer: The references have been added to the manuscript
  2. Query: The table 1 and table 2 provides least information. The authors should consider adding more data on drug efficacy values, side effects and references. Answer: Table 3 showing characteristics and results from main clinical studies on JAKinibs in axSpA has been added to the manuscript. Figure 2 and 3 have been also added to provide more information on the point of action of JAKinibs in axSpa and a flow chart for selecting biologics and JAKinibs for axSpA treatment.
  3. Query: The clinical trials so far conducted on using JAK Inhibitors can also be depicted in a graphical or tabular format. Answer: These data were reported in Table 3, added to the manuscript.

Reviewer 2 Report

This paper is well written, and the text is clear and easy to read. However, careless mistakes are scattered throughout, so the manuscript needs to be reviewed and revised. Also, it's a pity that the tables and figure only have general contents of JAK inhibitors. 

1. Please add keywords. 

2. Since there is a description of TNFα in part, please unify it with TNF-α.

3. Some font sizes are incorrect.

4. In Table 1, correct "RA =" to the appropriate position.

5. Please change Not to No in Table 1. 

6. It would be nice to have a graphic on axSpA-specific pathology and point of action of JAK inhibitors.

7. It would be useful for the reader to provide a flow chart for selecting biologics and JAK inhibitors for patients with axSpA. 

Author Response

Reviewer #2

  1. Query: Please add keywords. Answer: Keywords have been added
  2. Query: Since there is a description of TNFα in part, please unify it with TNF-α. Answer: The abbreviation has been expressed as TNFα throughout the text
  3. Query: Some font sizes are incorrect. Answer: Wrong font sizes have been corrected
  4. Query: In Table 1, correct "RA =" to the appropriate position. Answer: Table 1 has been checked and corrected
  5. Query: Please change Not to No in Table 1. Answer: Not has been changed to No
  6. Query: It would be nice to have a graphic on axSpA-specific pathology and point of action of JAK inhibitors. Answer: Figure 2 has been added showing immunopathology and point of action of JAKinibs
  7. Query: It would be useful for the reader to provide a flow chart for selecting biologics and JAK inhibitors for patients with axSpA. Answer: Fig. 3 showing a flow chart for selecting biologics and JAKinibs for patients with axSpa has been added, in accordance with ASAS-EULAR recommendations. A reference has been added in this regard.
  8. Query: Careless mistakes are scattered throughout, so the manuscript needs to be reviewed and revised. Answer: Manuscript has been reviewed and mistakes has been corrected.

9: Query: it's a pity that the tables and figure only have general contents of JAK inhibitors. Answer: We have added Table 3 and Figures 2 and 3 to provide more details on the use of JAK inhibitors in the treatment of axSpA.

Round 2

Reviewer 1 Report

Thank you for the corrections. 

Author Response

Thank you for the fundamental suggestions needed to improve our manuscript

Reviewer 2 Report

The content of Figure 2 is difficult to understand. Please write the legend properly.

Author Response

Query: The content of Figure 2 is difficult to understand. Please write the legend correctly. Answer: The legend in Figure 2 has been entirely rewritten (highlighted in green) for detailed explanation and to improve its clarity